# Influence of the Polarity of the Plasticizer on the Mechanical Stability of the Filler Network by Simultaneous Mechanical and Dielectric Analysis

**DOI:** 10.3390/polym14102126

**Published:** 2022-05-23

**Authors:** Sahbi Aloui, Horst Deckmann, Jürgen Trimbach, Jorge Lacayo-Pineda

**Affiliations:** 1NETZSCH-Gerätebau GmbH, Schulstraße 6, 29693 Ahlden, Germany; horst.deckmann@netzsch.com; 2Hansen & Rosenthal KG, Am Sandtorkai 64, 20457 Hamburg, Germany; juergen.trimbach@hur.com; 3Continental Reifen Deutschland GmbH, Jädekamp 30, 30419 Hannover, Germany; jorge.lacayo-pineda@conti.de

**Keywords:** plasticizer, polarity, carbon black network, simultaneous mechanical and dielectric analysis, mechanical stability

## Abstract

Four styrene butadiene rubber (SBR) compounds were prepared to investigate the influence of the plasticizer polarity on the mechanical stability of the filler network using simultaneous mechanical and dielectric analysis. One compound was prepared without plasticizer and serves as a reference. The other three compounds were expanded with different plasticizers that have different polarities. Compared with an SBR sample without plasticizer, the conductivity of mechanically unloaded oil-extended SBR samples decreases by an order of magnitude. The polarity of the plasticizer shows hardly any influence because the plasticizers only affect the distribution of the filler clusters. Under static load, the dielectric properties seem to be oil-dependent. However, this behavior also results from the new distribution of the filler clusters caused by the mechanical damage and supported by the polarity grade of the plasticizer used. The Cole–Cole equation affirms these observations. The Cole–Cole relaxation time τ  and thus, the position of maximal dielectric loss increases as the polarity of the plasticizer used is also increased. This, in turn, decreases the broadness parameter α implying a broader response function.

## 1. Introduction

Plasticizers are a widely used additive in rubber compounds [1,2,3,4]. They are particularly important and, as the third-highest ingredient in terms of content level, come in right after rubber and fillers. As processing aids, the plasticizers are added in different concentrations in order to impart rubber products with the desired elastic properties in the operating temperature range [5,6,7,8,9,10].

As a fluid component, the plasticizer migrates in the rubber matrix and its macromolecules are integrated into the polymer chains through intermolecular interactions. Consequently, the intermolecular forces of the polymer chains and the number of free valences in the three-dimensional structure are reduced. The internal space between the polymer chains is thus larger, and the free volume that allows the polymer chains to flow above their glass transition temperature increases [11,12,13,14,15]. This new conformation of the polymer chains, in turn, increases their mobility and enhances the filler distribution in the rubber mixture [16,17,18,19,20,21]. Above a certain percolation threshold, a filler network is formed that reinforces the rubber compounds and provides the necessary mechanical stability [16,17]. This applies to both the carbon-based fillers such as carbon black and silica [18,19,20,21]. Indeed, the plasticizer type strongly affects the mechanical properties of rubber products due to a shift in the glass transition temperature. Consequently, the strain, the mechanical stress, the modulus of elasticity and the damping behavior change [22,23,24].

Furthermore, the dielectric properties of rubber samples filled with electrically conductive filler depend on the structure of its filler network [25,26,27,28,29,30,31,32]. This applies to filler networks made of electrically conductive fillers such as carbon-based carbon black or hybrid filler networks, provided that at least one electrically conductive filler is present [25,26,27,28]. The non-conductive component is mainly used because of its excellent mechanical reinforcement, as is the case with silica used in dynamic systems such as car tires [29,30,31,32]. Aloui et al. have shown that mechanically induced changes in the structure of the electrically conductive filler network have a direct impact on dielectric mechanisms such as charge transport and polarization [33,34]. These, in turn, have consequences for the dielectric constant and the dielectric conductivity of rubber samples [35,36,37,38,39].

The direct relationship between mechanical and dielectric properties makes simultaneous mechanical and dielectric analysis of rubber samples filled with electrically conductive filler an outstanding technique for opening up new horizons in evaluating the microstructure dynamics of rubber materials under mechanical load and hence reproducing authentic situations from operation modes [40,41,42,43]. In addition to quality measurements on test samples, examinations on installed end products can also be guaranteed if sensors are installed to record the current material properties during use and to monitor them in the subsequent step. Mainly the dielectric properties are used as a response to the mechanical load [44].

In this study, the influence of the polarity of the plasticizer on the mechanical behavior of carbon black filled SBR under static loading is investigated. It is about the dynamics of the reinforcing filler network under mechanical loading and how this behavior can be described with simple material models. This study serves to understand the usage process and opens up the possibility of describing dynamic systems such as tires or seals during application. The electrical response is used as the display variable.

In this study, the influence of the plasticizer on the dielectric response of carbon black filled SBR under static load is examined, particularly with respect to the polarity of the plasticizer. The simultaneous dynamic-mechanical and dielectric analyzer DiPLEXOR^®^ 500 N from NETZSCH-Gerätebau in Ahlden, Germany is used for this purpose.

## 2. Excursus: Dielectric Relaxation in Elastomers

Dielectric relaxation describes the build-up of the electric polarization of a dielectric medium after application of an external electric field. The characterization of the dielectric relaxation is based on the measurement of the variation of the permittivity as a function of frequency. The permittivity stems from dipole orientation and transport of free charge carriers under the action of an electric field. The measuring method uses capacitance measurements as a function of frequency for a sample placed between two electrodes. An extensive explanation of the phenomenon and the measurement technology can be found in [45].

The permittivity $\varepsilon^{*}$ is a complex function with the real part $\varepsilon^{\prime }$ and the imaginary part $\;\varepsilon^{\prime \prime }$, also known as dielectric loss. As is typical for elastomers, not all dipoles have the same relaxation time, but different relaxation times, which exhibit a distribution with a relaxation peak. In order to describe these types of relaxation correctly, there are various empirical models derived from the Debye equation. In the case of symmetrical frequency response, the Cole–Cole approach is mainly used for amorphous dielectrics [46]. According to the Cole–Cole equation,
(1)$$\varepsilon^{*}\left(\omega \right)=\varepsilon_{\inf}+\frac{\Delta \varepsilon }{1+{\left(i\omega \tau \right)}^{\alpha }}\;\mathrm{with}\;0<\alpha \leq 1$$
where $\varepsilon_{\inf}\;$ is the infinite frequency dielectric permittivity, Δε is the relaxation strength, α is the broadness parameter and τ is the Cole–Cole relaxation time. $\omega =2\pi f_{\mathrm{el}}$ is the angular frequency and $f_{\mathrm{el}}$ is the electrical frequency. The expressions of $\varepsilon^{\prime }$ and $\varepsilon^{\prime \prime }$ take the following form:(2)$$\varepsilon^{\prime }\left(\omega \right)=\varepsilon_{\inf}+\Delta \varepsilon \cdot \frac{1+{\left(\omega \tau \right)}^{\alpha }\cos\left[\alpha \frac{\pi }{2}\right]}{1+2{\left(\omega \tau \right)}^{\alpha }\cos\left[\alpha \frac{\pi }{2}\right]+{\left(\omega \tau \right)}^{2\alpha }}$$

And
(3)$$\varepsilon^{\prime \prime }\left(\omega \right)=\frac{\sigma_{dc}}{\omega \varepsilon_{0}}+\Delta \varepsilon \cdot \frac{{\left(\omega \tau \right)}^{\alpha }\sin\left[\alpha \frac{\pi }{2}\right]}{1+2{\left(\omega \tau \right)}^{\alpha }\cos\left[\alpha \frac{\pi }{2}\right]+{\left(\omega \tau \right)}^{2\alpha }}$$
where $\;\sigma_{dc}$ is the direct current conductivity or DC conductivity [33,34].

## 3. Materials

Four carbon black filled SBR based compounds were prepared at Hansen and Rosenthal KG in Hamburg, Germany. The carbon black N 330 was used at a filler concentration of 60 phr. For a reference sample, no plasticizer was added. The three other samples each contain 20 phr of one plasticizer grade, which differ by polarity. Of course, the good miscibility of the plasticizers in the rubber matrix must be taken into account. Therefore, the following plasticizers are used: The plasticizers used are a paraffinic base oil (SN400), mild extraction solvate (MES) and distillate aromatic extract (DAE). Figure 1 shows the structural formula of the plasticizers with different polarities used [47].

Plasticizers SN400, MES and DAE have an aniline point in accordance with DIN ISO 2977 at 101 °C, 84 °C and 43 °C [47]. The aniline point is the temperature at which a homogeneous mixture of equal volumes of aniline and plasticizer separates into 2 phases during the cooling process. The degree of miscibility of aniline with the plasticizer estimates the aromatic content in the plasticizer. The lower the aniline point, the more polar the plasticizer.

The solubility parameter δ is an indicator of the miscibility quality of the various plasticizers within the SBR matrix. SN400, MES and DAE have a solubility parameter δ of 16 MPa^1/2^, 16.7 Mpa^1/2^ and 18.5 Mpa^1/2^. With a value of 17.2 Mpa^1/2^, the solubility parameter δ for SBR is in the same range as for the plasticizers, implying a good compatibility [47].

The aniline point and the solubility parameter are shown in Figure 2.

The compound formulation is shown in Table 1.

## 4. Methods

### 4.1. Dielectric Analysis

Purely dielectric measurements were carried out at room temperature using the broadband dielectric spectrometer BDS from Novocontrol in Montabaur, Germany. The mechanical load was infinitesimally small, and it only served to maintain contact between the SBR samples and the electrodes. The coin-shape samples had a diameter of 30 mm and a thickness of 0.1 to 0.3 mm. The applied sinusoidal alternating voltage had an amplitude of 3V. The electrical frequency ranged between 1 Hz and 1 MHz.

### 4.2. Simultaneous Mechanical and Dielectric Analysis

Simultaneous mechanical and dielectric analysis were performed on the DiPLEXOR 500 N of NETZSCH-Gerätebau in Ahlden, Germany. The DiPLEXOR 500 N is the result of coupling the EPLEXOR 500 N dynamic-mechanical analyzer with the broadband dielectric spectrometer BDS from Novocontrol in Montabaur, Germany. Coin-shaped samples with a diameter of 10 mm and thickness of around 2 mm were used. The measurements were carried out at room temperature applying a static force of 10 N and a sinusoidal alternating voltage with an amplitude of 1 V. The electrical frequency ranged between 1 Hz and 1 MHz.

Three measurements were performed to confirm the validity of the results. However, only one measurement is shown to represent the overall result.

## 5. Results and Discussion

Carbon black filled elastomers have permanent dipoles and free charge carriers on the surface area of the carbon black clusters due to the graphitized surface area of carbon black particles.

The physical mechanisms behind the dielectric response of carbon black filled elastomers are based on the electrical frequency of the electrical alternating field applied. In the frequency range between 1 Hz and 1 MHz, the free charge carriers can be transported along the electric field lines. This conduction mechanism, described by the dielectric conductivity $\;\sigma^{*}$, is initially frequency-independent and reaches a constant plateau value, known as direct current conductivity or DC conductivity, abbreviated $\;\sigma_{dc}$. This is the result of phase-equal change in the electric field and the sample polarization. From a material-dependent frequency threshold, $\sigma^{*}$ becomes frequency-dependent because the change in electric field and the change in sample polarization become time-delayed. This dielectric dispersion is caused by additional relaxation processes which come into play. This part is known as AC conductivity.

Furthermore, the present dipoles in the SBR materials are oriented along the electric field lines. Orientation polarization arises. Depending on the sample thickness, the applied electric field can also lead to accumulation of dipoles at the interfaces, also known as interface polarization. Both polarization mechanisms are described by the permittivity $\;\varepsilon^{*}$.

### 5.1. Dielectric Analysis

Purely dielectric measurements on the SBR samples are performed at room temperature without mechanical load. This serves first to determine the contribution of the different components to the dielectric response without the influence of static load. The measurements are performed on SBR samples with a thickness of 0.1 to 0.3 mm. Figure 3 shows the frequency-dependent change in the real part of the conductivity $\sigma^{\prime }$ of the SBR samples with and without plasticizers.

Figure 3 indicates that the real part of the conductivity $\sigma^{\prime }$ of the SBR sample without plasticizer hardly shows any changes within the frequency range of the measurements. A direct current conductivity $\;\sigma_{dc}$ of 4.8 × 10^−4^ S/cm is recorded. It is to note that this high conductivity in carbon black filled rubber is clearly related to the carbon black network. At a filler concentration of 60 phr, the mechanical percolation threshold is exceeded, resulting in a network of interconnected filler clusters. The contribution of polymer chains to the conductivity of SBR samples is approximately ten orders of magnitude less.

The oil-extended SBR samples show at least two disparities compared with the SBR sample without plasticizer. First, the real part of the conductivity $\sigma^{\prime }$ is from an electrical frequency of 100 kHz frequency-dependent, regardless of the polarity of the plasticizer used. $\sigma^{\prime }$ increases with increasing frequency. Second, the direct current conductivity $\;\sigma_{dc}$ becomes smaller. It is 7 × 10^−5^ S/cm for the SBR samples with 20 phr SN400 and DAE. For the SBR samples with 20 phr MES, $\;\sigma_{dc}$ is 5.8 × 10^−5^ S/cm.

In this context, it is worthy to note that the polarity of the plasticizer has no big influence on the conductivity. The addition of plasticizer has only influenced the distribution of the filler clusters within the rubber matrix. As a result, the distances covered by the free charge carriers along the electric field lines become longer.

### 5.2. Simultaneous Mechanical and Dielectric Analysis

Dielectric measurements on the SBR samples are carried out at room temperature under a static force of 10 N. The latter corresponds to a mechanical stress of 0.127 MPa. Figure 4 shows the frequency-dependent change in the real part of the conductivity $\sigma^{\prime }$ of the SBR samples with and without plasticizers.

Figure 4 shows a completely different picture than Figure 3. This has to do with the different geometries of the test specimens and the applied contact forces. The simultaneous mechanical and dielectric analyses were performed on 2-mm-thick SBR samples with a contact force of 10 N. The purely dielectric measurements were performed on SBR samples with a thickness of 0.1 to 0.3 mm and a contact force in the mN range. It is therefore not possible to readily perform a direct comparison of the absolute measured values.

The real part of the conductivity $\sigma^{\prime }$ becomes frequency-dependent for all SBR samples independently of the plasticizer content. The characteristic frequency between the DC and AC conductivity shifts towards lower frequencies with the addition of the plasticizer.

For the SBR sample without plasticizer, the application of a static force of 10 N reduces $\;\sigma_{dc}$ from 4.8 × 10^−4^ S/cm to 1.4 × 10^−4^ S/cm. This is due to the mechanical damage that reduces the density of the conduction paths within the carbon black network, and thus the SBR samples.

For the SBR samples with plasticizer, the polarity of the plasticizer used strongly influences the conductivity. $\;\sigma_{dc}$ is 10^−4^ S/cm for the SBR sample with 20 phr SN400, 7.7 × 10^−5^ S/cm for the SBR sample with 20 phr MES and 2.7 × 10^−5^ S/cm for the SBR sample with 20 phr DAE. It is obvious that increasing the polarity of the plasticizer used decreases $\;\sigma_{dc}$.

The AC conductivity, abbreviated $\;\sigma_{ac}$, is the frequency-dependent part of $\;\sigma^{\prime }$ at which the change in the electric field and the change in sample polarization are time-delayed. The curve shape of $\;\sigma_{ac}$ for the different SBR samples also suggests that the polarity of the plasticizer used has a huge influence on the relaxation processes caused by this dielectric dispersion.

However, as mentioned in the previous section, the polarity has no influence on the conductivity. It strongly affects the distribution of the filler clusters within the rubber matrix.

The difference in DC conductivity $\;\sigma_{dc}$ between the mechanically undamaged (without mechanical load) and damaged (static force of 10 N) SBR samples is illustrated in Figure 5.

Figure 5 shows that $\;\sigma_{dc}$ of the undamaged SBR samples is only affected by the presence of plasticizer, and not by the polarity of the plasticizer used. In contrast, $\;\sigma_{dc}$ of the mechanically loaded and hence damaged SBR samples seems to be indirectly influenced by the polarity of the plasticizer used. In fact, the use of plasticizer influences the inner structure of the SBR samples by generating new dispersion states following the mechanical load.

In order to examine the impact of the plasticizer and its polarity on the dispersive part of the conductivity $\;\sigma_{ac}$, it is more convenient to consider the global dielectric response of the SBR samples, expressed in terms of permittivity. In addition to conductivity, relaxation processes are taken into account. Figure 6 and Figure 7 show the change in the real and imaginary part of the permittivity, $\varepsilon^{\prime }$ and $\varepsilon^{\prime \prime }$ of SBR samples with different plasticizers at 24 °C under a static force of 10 N.

### 5.3. Cole–Cole Relaxation Model

To determine the dielectric relaxation in the SBR samples, the real and imaginary part of the permittivity $\varepsilon^{\prime }$ and $\varepsilon^{\prime \prime }$ are simultaneously fitted with the Equations (2) and (3) according to the Cole–Cole approach. With these basic equations, an own fitting program was developed, which simultaneously fits the real and imaginary part of the permittivity. Figure 8 displays the behavior of $\varepsilon^{\prime }\;$ and $\;\varepsilon^{\prime \prime }\;$ at 24 °C under a static force of 10 N. The SBR sample filled with 20 phr MES is shown as representative for all other SBR samples.

Figure 8 suggests a good fitting quality for the measurement data of the SBR sample filled with 20 phr MES at 24 °C under a static force of 10 N. The fitting parameters are displayed on the inset. Hereinafter, the individual fitting parameters related to all SBR samples are presented. Figure 9 first shows the Cole–Cole relaxation time  τ.

The Cole–Cole relaxation time τ gives the position of maximal dielectric loss. Figure 9 shows that τ increases as the polarity of the plasticizer used increases. τ is related to a characteristic electrical frequency of maximal loss $f_{\mathrm{el}}$ according to $\tau =1/\left(2\pi f_{\mathrm{el}}\right)$. Increasing τ means a decrease in the characteristic electrical frequency of maximal loss $\;f_{\mathrm{el}}$. This resembles the results shown in Figure 4, in which the frequency limit between DC and AC conductivity shifts to lower frequencies with higher polarity.

In the following, Figure 10 illustrates the broadness parameter  α  of the SBR samples determined by means of simultaneous mechanical and dielectric measurements at 24 °C under a static force of 10 N.

In contrast to the Debye model where the broadness parameter  α equals one, Figure 10 shows that all α  values for SBR samples are less than one. This indicates a symmetrically broad loss peak. Furthermore, α decreases as the polarity of the plasticizer used is increased, implying a broader response function. This behavior is caused by the interaction of the dipoles with each other, inducing a dispersion of the relaxation time  τ. In physical terms, the dipoles participating in the relaxation phenomenon do not have the same relaxation time. Most probably this behavior is promoted by the polarity of the plasticizer, since the more polar the plasticizer, the more dipoles are available.

Figure 11 depicts the infinite-frequency dielectric permittivity $\;\varepsilon_{\inf}\;$ of the SBR samples determined by means of simultaneous mechanical and dielectric measurements at 24 °C under a static force of 10 N. $\varepsilon_{\inf}=\underset{\omega \to \infty }{\lim}\varepsilon^{\prime }\left(\omega \right)$ is also known as the high frequency permittivity.

Figure 11 shows that $\;\varepsilon_{\inf}$ seems to depend on the presence of plasticizer within the SBR samples rather than the type of plasticizer. The doubling of the high frequency permittivity $\;\varepsilon_{\inf}$ for the oil-extended SBR samples as compared to the SBR sample without plasticizer can be attributed to the increase in dead ends caused by the mechanical damage of the filler network and the presence of additional polar molecules. Locally, more polar regions arise and permittivity increases.

The last parameter according to the Cole–Cole Equation (1) is the relaxation strength ∆ε and it is shown in Figure 12.

The relaxation strength  Δε shown in Equation (1) is the difference between the static dielectric permittivity $\varepsilon_{S}=\underset{\omega \to 0}{\lim}\varepsilon^{\prime }\left(\omega \right)$, also known as low frequency permittivity, and $\varepsilon_{\inf}$. $\Delta \varepsilon =\varepsilon_{S}-\varepsilon_{\inf}$ gives the contribution of the orientation polarization to the dielectric function and evaluates the mean molecular dipole moment on the conditions that the present dipoles do not interact with each other and shielding effects are insignificant [19]. Since this is not the case for the SBR samples, no reliable conclusions can be drawn in this regard.

## 6. Summary and Conclusions

Four SBR compounds were prepared to investigate the influence of plasticizer polarity on the mechanical stability of the filler network using simultaneous mechanical and dielectric analysis.

The SBR compound without plasticizer serves as a reference. In a mechanically unloaded state, a constant conductivity of 4.8 × 10^−4^ S/cm was measured, showing no influence exerted by the electrical frequency. Under a static force of 10 N, which corresponds to a mechanical load of 0.127 MPa, the conductivity becomes frequency-dependent. The direct current conductivity $\sigma_{dc}$ decreases from 4.8 × 10^−4^ S/cm to 1.4 × 10^−4^ S/cm due to the generated mechanical damage that reduces the density of the conduction paths within the carbon black network, and thus the SBR samples.

The other three compounds are expanded with oils that have different polarities.

The picture looks different for the three oil-expanded SBR samples. From an electrical frequency of 100 kHz, the real part of the conductivity $\sigma^{\prime }$ shows a dispersive part and increases as the frequency increases, regardless of the polarity of the plasticizer used. In addition, $\sigma_{dc}$ is lower by almost an order of magnitude. It is 7 × 10^−5^ S/cm for the SBR samples with 20 phr SN400 and DAE. For the SBR samples with 20 phr MES, $\;\sigma_{dc}$ is 5.8 × 10^−5^ S/cm.

The polarity of the plasticizers used does not directly contribute to the conductivity of the SBR samples. It solely influences the distribution of the filler particles within the matrix by increasing the distances covered by the free charge carriers along the electric field lines.

Under mechanical stress, the conductivity of all SBR samples becomes frequency-dependent, independent of the plasticizer content and type. Furthermore, applying a static force of 10 N reduces $\;\sigma_{dc}$ from 4.8 × 10^−4^ S/cm to 1.4 × 10^−4^ S/cm for the SBR sample without plasticizer. This also applies to the SBR sample with 20 phr DAE. $\;\sigma_{dc}$ decreases from 7 × 10^−5^ S/cm to 2.7 × 10^−5^ S/cm. For the SBR samples with 20 phr SN400 and MES, a small increase in $\;\sigma_{dc}$ is observed. $\;\sigma_{dc}$ is 10^−4^ S/cm for the SBR sample with 20 phr SN400 and 7.7 × 10^−5^ S/cm for the SBR sample with 20 phr MES. These experimental findings can be explained by the new internal structure, which is characterized not only by damage to the filler clusters, but also by a new distribution of the filler clusters within the matrix. It is also worth noting that $\;\sigma_{dc}$ for the oil-expanded SBR samples decreases as polarity of the plasticizer increases.

The dielectric relaxation was analyzed to describe the behavior of the SBR samples under mechanical stress. The real and imaginary part of the permittivity, $\varepsilon^{\prime }$ and $\varepsilon^{\prime \prime }$, were simultaneously fitted according to the Cole–Cole approach. The Cole–Cole relaxation time τ  and thus also the position of maximal dielectric loss both increase as the polarity of the plasticizer used increases. This, in turn, decreases the broadness parameter α, implying a broader response function. The high frequency permittivity $\;\varepsilon_{\inf}$ depends on the presence of plasticizer within the SBR samples rather than the type of plasticizer. $\varepsilon_{\inf}$ for the oil-expanded SBR samples is double that of the SBR sample without plasticizer. This can be attributed to the increase in dead ends, implying more polar regions in the SBR matrix. Finally, no clear trend can be seen for the relaxation strength  Δε. No reliable conclusions can be drawn in this regard.

This study is primarily application-oriented. These investigations are intended to describe and evaluate the influence of mechanical stress on a rubber compound in use as simply as possible. Rubber mixtures are mechanically loaded with the dynamic-mechanical analyzer in order to transfer the real operating conditions from the field to the laboratory and then to characterize the material behavior based on the dielectric properties.

In future work, the mechanical loads will be increased in order to reach the non-linear range of the rubber compounds in order to create conditions that are as realistic as possible. The study of rubber compounds with hybrid filler systems is also planned in order to characterize end products such as tires while driving, since the tire treads usually have a carbon black/silica filler system.

## Figures and Tables

**Figure 1 polymers-14-02126-f001:**
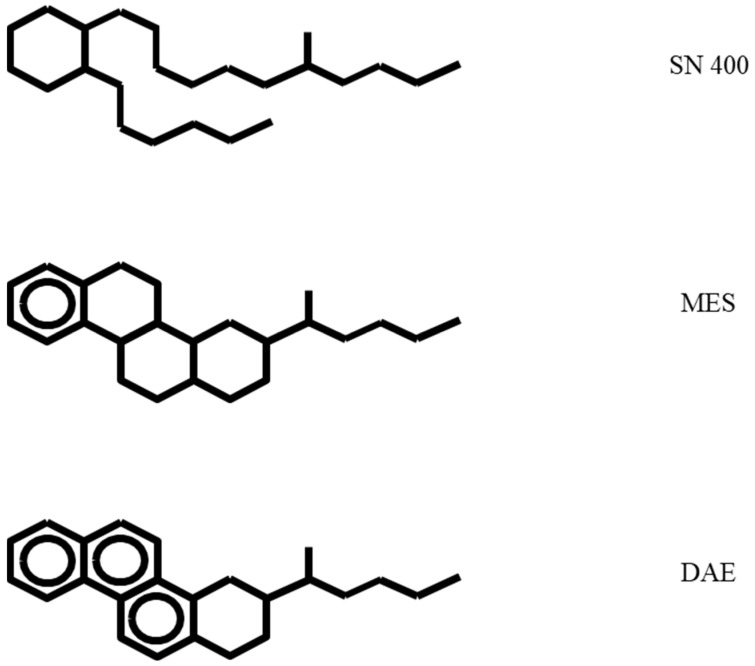
Structural formula of plasticizers SN400, MES and DAE.

**Figure 2 polymers-14-02126-f002:**
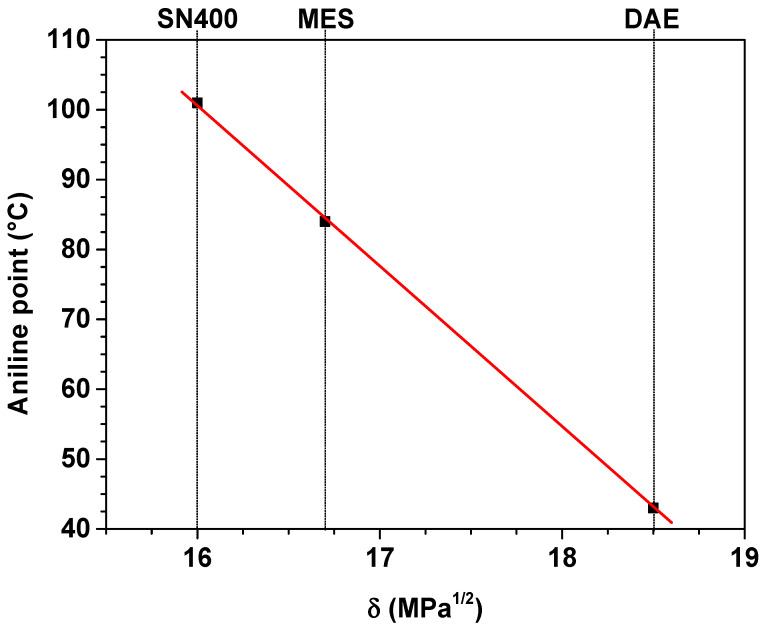
Aniline point and solubility parameter of the plasticizers.

**Figure 3 polymers-14-02126-f003:**
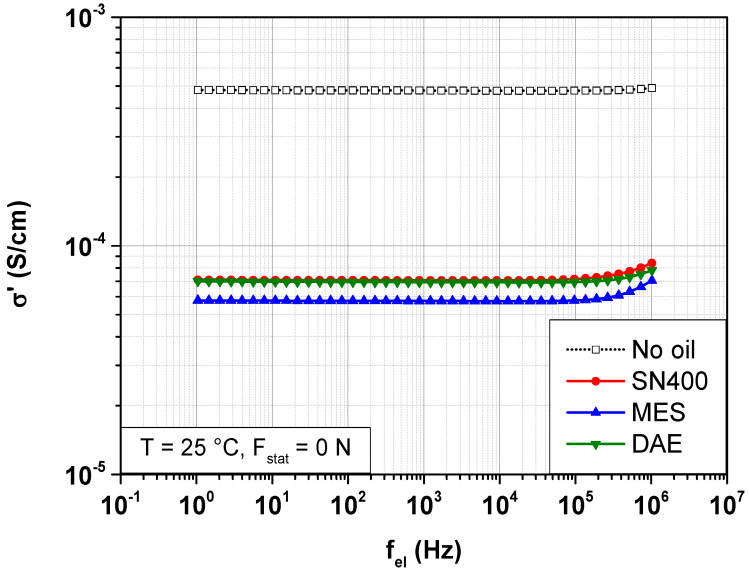
Frequency-dependent change of the real part of the conductivity $\sigma^{\prime }$ of SBR samples with different plasticizers at 25 °C.

**Figure 4 polymers-14-02126-f004:**
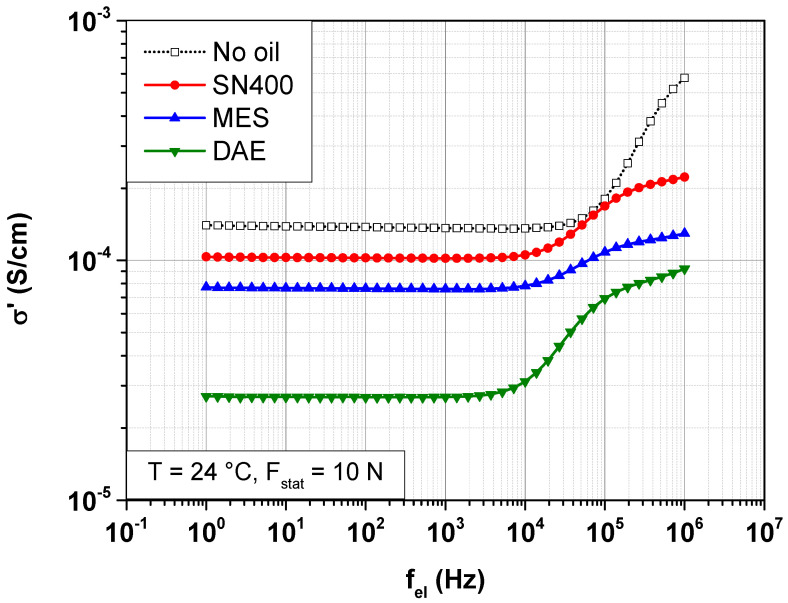
Frequency-dependent change in the real part of the conductivity $\sigma^{\prime }$ of SBR samples with different plasticizers at 24 °C under a static force of 10 N.

**Figure 5 polymers-14-02126-f005:**
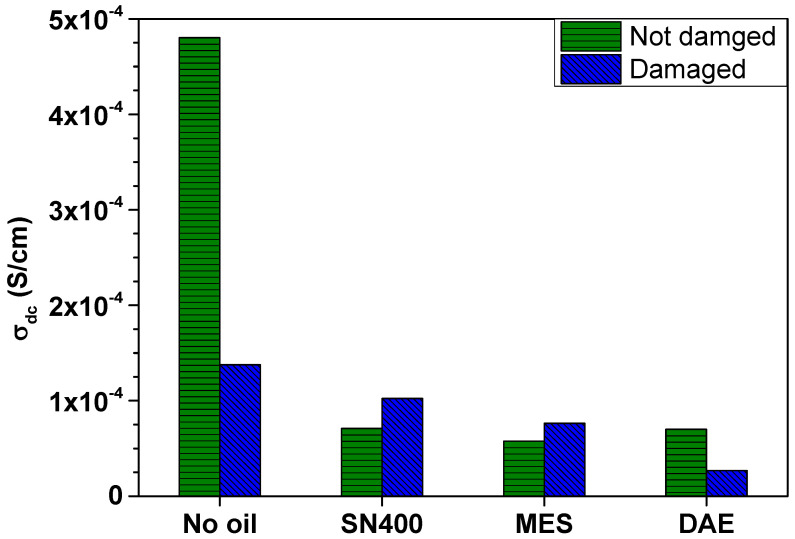
Difference in DC conductivity $\;\sigma_{dc}$ between the mechanically not damaged (without static load) and damaged (static force of 10 N) SBR samples.

**Figure 6 polymers-14-02126-f006:**
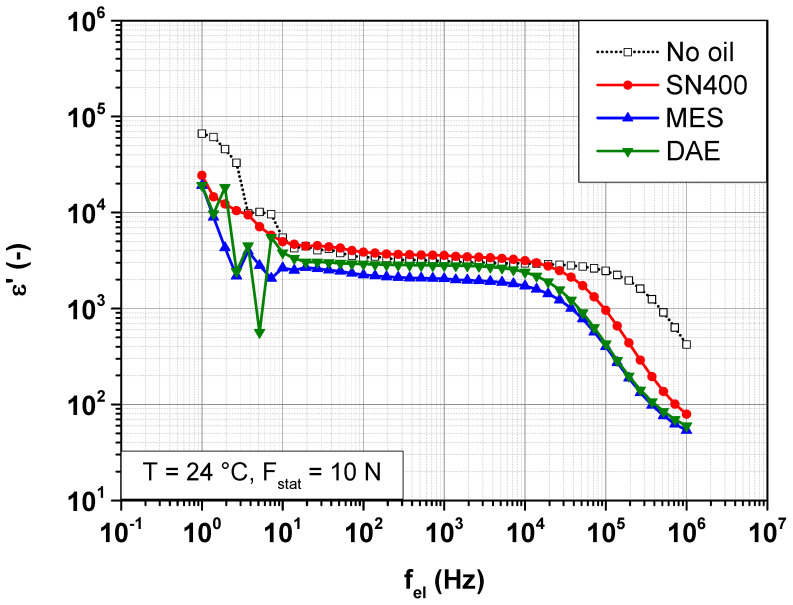
Frequency-dependent change in the real part of the permittivity $\varepsilon^{\prime }$ of SBR samples with different plasticizers at 24 °C under a static force of 10 N.

**Figure 7 polymers-14-02126-f007:**
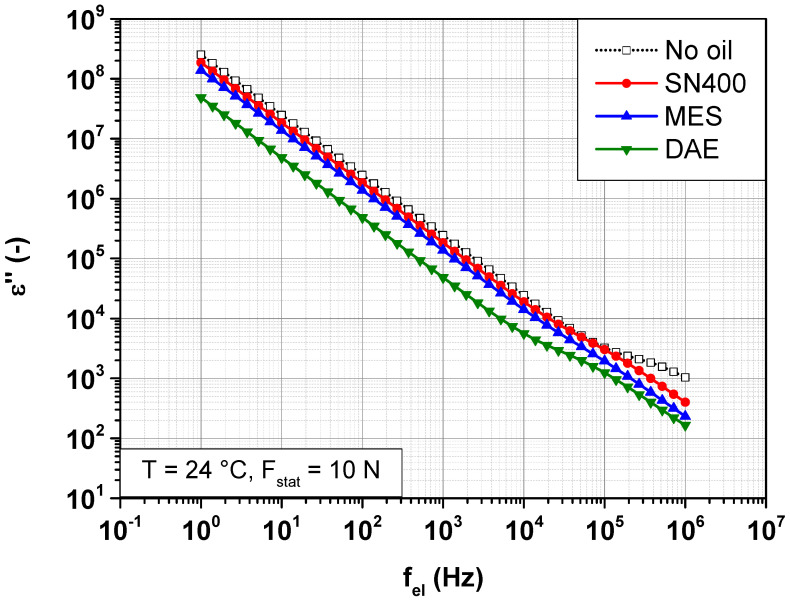
Frequency-dependent change in the imaginary part of the permittivity $\varepsilon^{\prime \prime }$ of SBR samples with different plasticizers at 24 °C under a static force of 10 N.

**Figure 8 polymers-14-02126-f008:**
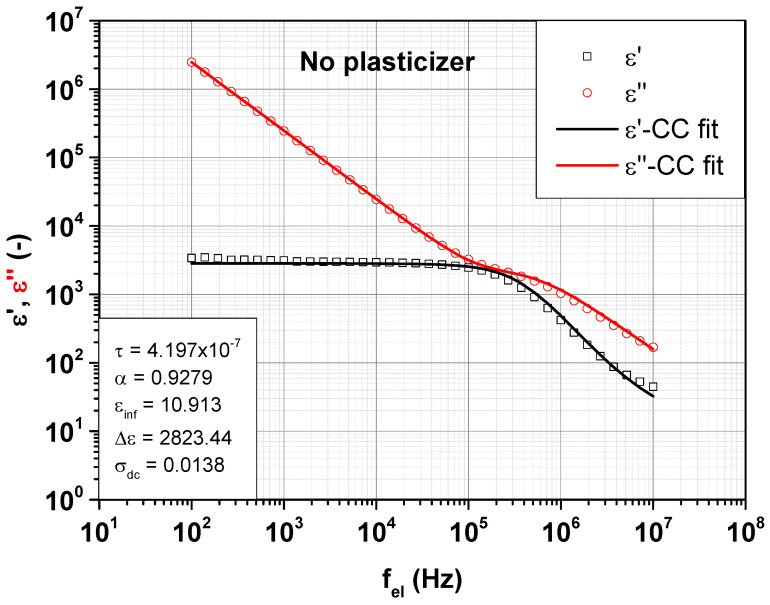
Simultaneous Cole–Cole fitting of real and imaginary part of the permittivity $\;\varepsilon^{\prime }\;$ and $\;\varepsilon^{\prime \prime }\;$ of SBR sample filled with 20 phr MES at 24 °C under a static force of 10 N.

**Figure 9 polymers-14-02126-f009:**
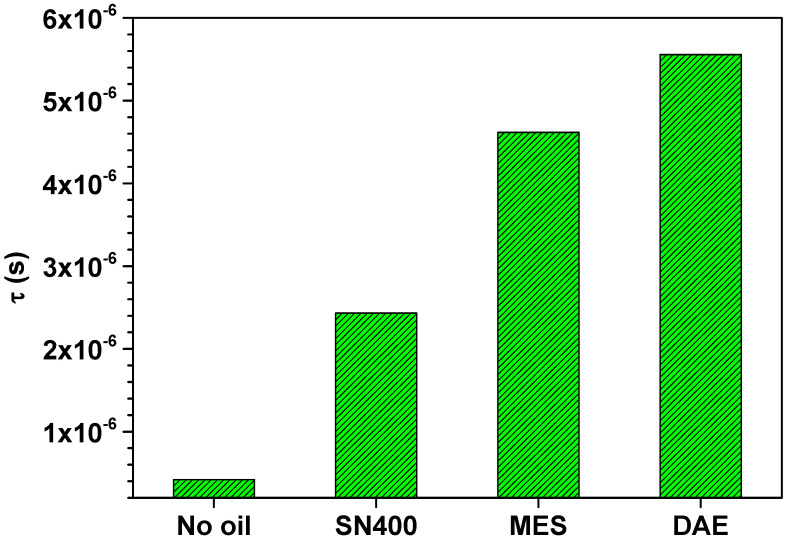
Cole–Cole relaxation time  τ of the SBR samples determined by means of simultaneous mechanical and dielectric measurements at 24 °C under a static force of 10 N.

**Figure 10 polymers-14-02126-f010:**
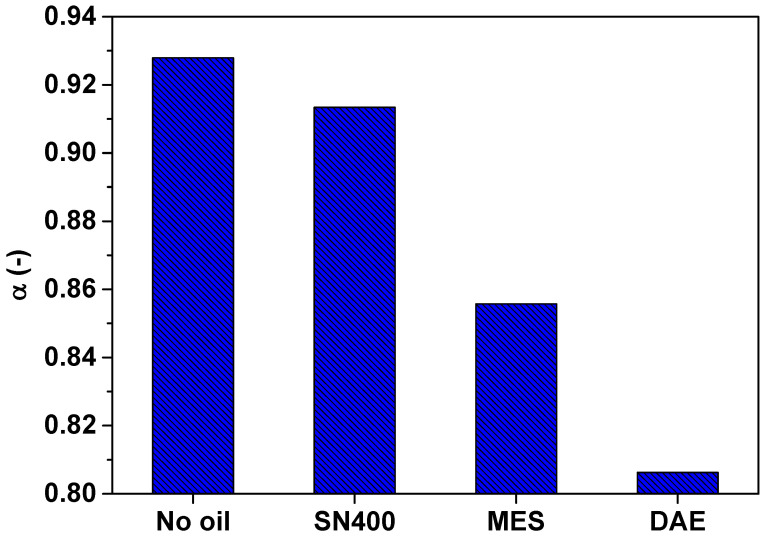
Broadness parameter  α  of the SBR samples determined by means of simultaneous mechanical and dielectric measurements at 24 °C under a static force of 10 N.

**Figure 11 polymers-14-02126-f011:**
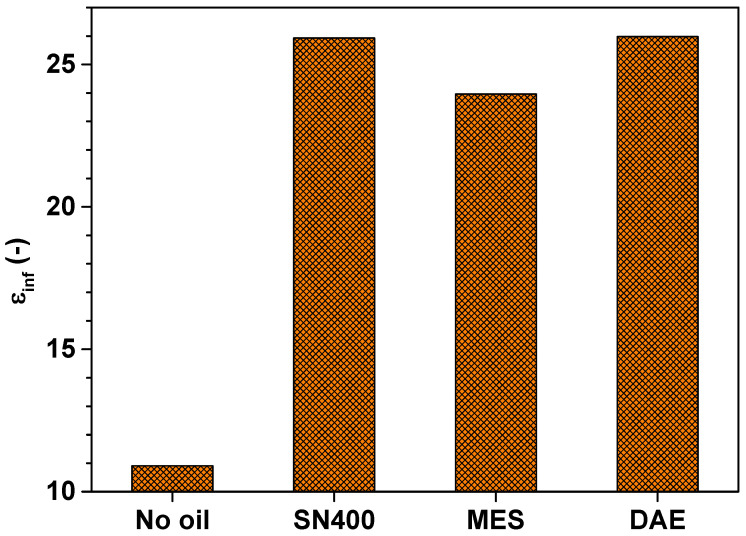
Infinite-frequency dielectric permittivity $\;\varepsilon_{\inf}\;$ of the SBR samples determined by means of simultaneous mechanical and dielectric measurements at 24 °C under a static force of 10 N.

**Figure 12 polymers-14-02126-f012:**
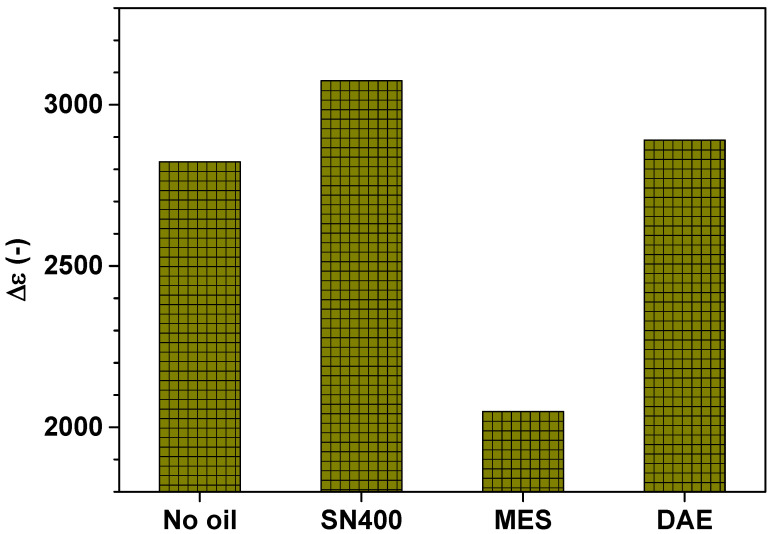
Relaxation strength  Δε  of the SBR samples determined by means of simultaneous mechanical and dielectric measurements at 24 °C under a static force of 10 N.

**Table 1 polymers-14-02126-t001:** Compound formulation in phr.

	No Oil	SN400	MES	DAE
SBR 1502	100	100	100	100
N 330	60	60	60	60
**SN400**	-	**20**	-	-
**MES**	-	-	**20**	-
**DAE**	-	-	-	**20**
ZnO	2.5	2.5	2.5	2.5
Stearic acid	1	1	1	1
TMQ	1	1	1	1
6PPD	1	1	1	1
CBS	1.8	1.8	1.8	1.8
DDTD	0.2	0.2	0.2	0.2
Sulphur	1.5	1.5	1.5	1.5

The antioxidants 2,2,4-Trimethyl-1,2-dihydrochinolin (TMQ) and N-(1.3-Dimethylbutyl)-N′-phenyl-p-phenylenediamine (6PPD) were added at a concentration of 1 phr. The samples were sulfur-vulcanized. In addition to sulfur, the vulcanization accelerators N-cyclohexyl-2-benzothiazolesulfenamide (CBS) and Dimethyldiphenylthiuram disulfide (DDTD) were used.

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
