# Peer review of "Influence of the Polarity of the Plasticizer on the Mechanical Stability of the Filler Network by Simultaneous Mechanical and Dielectric Analysis"

_polymers, 2022, doi:10.3390/polym14102126_

Round 1

Reviewer 1 Report

This paper presents an analysis of the influence of plasticizer polarity on some properties of SBR.

The authors can consider the following aspects:

- The paper may be of interest to the scientific community;

- No keywords specified!

- The Introduction section needs to be substantially improved, taking into account other scientific paper in the field. References of the type [16-21] and [25-32], respectively, should be avoided. Also at the end of this section, the objectives of the research and the structure of the paper should be presented more clearly.

- The research methodology must be detailed in order to explain the decision to consider SN 400, MES respectively DAE 20;

- A more detailed presentation of the initial properties of the materials used in the research is required;

- It is necessary to present macroscopic or even microscopic images of the samples used in research;

- The discussion part must be much developed in order to be able to highlight the novelty brought by the research presented in the paper in relation to other research in the field. In the current form, in the discussion part, we did not find that the results obtained were related to the results presented in other scientific papers;

- The conclusions should also present the practical applications of the results obtained and, at the end of them, the future research directions should be presented.

Author Response

Response to Reviewer 1 Comments

Point 1: The paper may be of interest to the scientific community.

Response 1: Thanks for your comment. We see that the paper is of interest to the scientific community and we hope you will too.

Point 2: No keywords specified!

Response 2: Thanks for your comment. Keywords have been added.

Point 3: The Introduction section needs to be substantially improved, taking into account other scientific paper in the field. References of the type [16-21] and [25-32], respectively, should be avoided. Also at the end of this section, the objectives of the research and the structure of the paper should be presented more clearly.

Response 3: Thanks for your comment. Section is amended accordingly.

Point 4: The research methodology must be detailed in order to explain the decision to consider SN 400, MES respectively DAE 20.

Response 4: Thanks for your comment. Keywords have been added

Point 5: A more detailed presentation of the initial properties of the materials used in the research is required.

Response 5: Thanks for your comment. All information provided are included in the manuscript.

Point 6: It is necessary to present macroscopic or even microscopic images of the samples used in research

Response 6: Thanks for the good remark. That would definitely be a plus. Unfortunately, we could not have taken microscopic pictures of the rubber compound. The samples are macroscopically indistinguishable.

Point 7: The discussion part must be much developed in order to be able to highlight the novelty brought by the research presented in the paper in relation to other research in the field. In the current form, in the discussion part, we did not find that the results obtained were related to the results presented in other scientific papers

Response 7: Thanks for the legitimate notice.

The idea of this work is original and application-oriented. These investigations are intended to describe and evaluate the influence of mechanical stress on a rubber compound in use as simply as possible. It is not about the analogy often found in earlier works between the mechanical and dielectric properties of rubber compounds. Therefore, we are not aware of any works that have the same intention as ours.

Point 8: The conclusions should also present the practical applications of the results obtained and, at the end of them, the future research directions should be presented.

Response 8: Thanks for your comment. Section is amended accordingly.

Reviewer 2 Report

The authors present a study on the influence of the plasticizer on the mechanical-dielectric coupling in carbon-black filled SBR material systems. The manuscript reports the issue of the mechanical stability of the filler network in three compounds prepared with different plasticizer polarities and fourth compound without plasticizer for reference. Conductivity results are presented after and upon stretching at room temperature.

The manuscript is in overall well written with a pretty good linguistic style. The study is very interesting but the reviewer has the following issues to address before to recommend the manuscript for publication:

  • The novelties of the paper should be clearly outlined in the last paragraph of the introduction section to justify the motivation for this study. It should be clearly highlighted how the proposed method will fill the gap in the existing literature. It is not very clear for a general reader.
  • Authors should enrich the literature by providing more references with respect to the present state of the art.
  • The methodology for the coupled mechanical and dielectric analysis is not clear. Can the authors present pictures of the used sample and the experimental protocol?
  • What is the stretch level during the mechanical experiments? Is there an effect of the stretch level on the conductivity results?
  • What about statistics? How many samples have been tested for each condition? Also, standard deviations can be inserted in the histograms in Figures 5, 9, 10, 11 and 12.
  • The fitting procedure must be described. The authors can indicate the software used to fit the parameters.
  • The manuscript presents several misprints that must be corrected.
  • Minor remark: It is not necessary to indicate the name of the dielectric analyzer in the introduction section.

Author Response

Response to Reviewer 2 Comments

Point 1: The novelties of the paper should be clearly outlined in the last paragraph of the introduction section to justify the motivation for this study. It should be clearly highlighted how the proposed method will fill the gap in the existing literature. It is not very clear for a general reader.

Response 1: Thanks for your comment. Section is amended accordingly.

Point 2: Authors should enrich the literature by providing more references with respect to the present state of the art.

Response 2: Thanks for the legitimate notice. The idea of this work is original and application-oriented. These investigations are intended to describe and evaluate the influence of mechanical stress on a rubber compound in use as simply as possible. It is not about the analogy often found in earlier works between the mechanical and dielectric properties of rubber compounds. Therefore, we are not aware of any works that have the same intention as ours.

Point 3: The methodology for the coupled mechanical and dielectric analysis is not clear. Can the authors present pictures of the used sample and the experimental protocol?

Response 3: Thanks for the comment. The geometry of the samples used for the dielectric analysis as well as the simultaneous mechanical and dielectric analysis is described in detail in Section 4.

Point 4: What is the stretch level during the mechanical experiments? Is there an effect of the stretch level on the conductivity results?

Response 4: Thanks for the comment. The mechanical properties of the rubber compounds are mentioned in Section 5.2. Of course, the degree of stretching affects the conductivity results. But that will be the subject of the study that is already planned: the influence of high mechanical loads, briefly introduced in the last paragraph in Conclusion.

Point 5: What about statistics? How many samples have been tested for each condition? Also, standard deviations can be inserted in the histograms in Figures 5, 9, 10, 11 and 12..

Response 5: Thanks for the legitimate question. The statistical approach was mentioned as the last sentence in Chapter 4.

Point 6: The fitting procedure must be described. The authors can indicate the software used to fit the parameters.

Response 6: Thanks for your comment. Section 5.3. is amended accordingly.

Point 7: The manuscript presents several misprints that must be corrected.

Response 7: Thanks for the good remark.

Point 8: Minor remark: It is not necessary to indicate the name of the dielectric analyzer in the introduction section.

Response 8: Thanks for your comment. Section is amended accordingly.

Round 2

Reviewer 1 Report

The authors revised their manuscript according to my suggestions. Thus the manuscript can be accepted for publication.

Reviewer 2 Report

The authors revised the article following reviewer comments. The article is in overall well written and it is worth publishing in Polymers. I recommend to accept the article in the present form.